

# Microbial processes in the weathering crust aquifer of a temperate glacier

Brent C. Christner[1,2*], Heather F. Lavender[2], Christina L. Davis[1], Erin E. Oliver[2‡], Sarah U. Neuhaus[3], Krista F. Myers[4], Birgit Hagedorn[5], Slawek M. Tulaczyk[3], Peter T. Doran[4], William C. Stone[6]

[1]University of Florida, Department of Microbiology and Cell Science, Biodiversity Institute, Gainesville, FL 32611 USA
[2]Louisiana State University, Department of Biological Sciences, Baton Rouge, LA 70803 USA
[3]University of California Santa Cruz, Department of Earth and Planetary Sciences, Santa Cruz, CA 95064 USA
[4]Louisiana State University, Department of Geology and Geophysics, Baton Rouge, LA 70803 USA
[5]Sustainable Earth Research LLC, Anchorage, AK 99508 USA
[6]Stone Aerospace, Del Valle, TX 78617 USA
[‡]Current address: San Diego State University, Department of Biology, San Diego, CA 92182 USA

*Correspondence to*: Brent C. Christner (xner@ufl.edu)

**Abstract.** Incident solar radiation absorbed within the ablation zone of glaciers generates a shallow perched aquifer and seasonal ice-bound microbial habitat. During the melt seasons of 2014 and 2015, borehole investigations were used to examine the physical, geochemical, and microbiological properties in the near-surface ice and aquifer of the temperate Matanuska Glacier (southcentral Alaska). Based on temperature, solar forcing, and ice optical properties, the dissipation of shortwave radiation promoted internal melting and the formation of a weathering crust with a maximum depth of ~2 m. Boreholes into the weathering crust provided access to water percolating through the porous ice. The water had low ion concentrations (4-12 $\mu$S cm$^{-1}$), was aerobic (12 mg $O_2$ L$^{-1}$), contained 200 to 8,300 cells mL$^{-1}$, and harbored growing populations with estimated in situ generation times of 11 to 14 days. During the melt season, the upper 2 m of ice experienced at least 3% of the surface photosynthetically active radiation flux and possessed a fractional water content as high as 10%. Photosynthetic subsistence of biogeochemical reactions in the weathering crust ecosystem was supported by ex situ metabolic experiments and the presence of phototrophic taxa (cyanobacteria, golden and green algae) in the aquifer samples. Melt water durations of ~7.5 months coupled with the growth estimates imply biomass may increase by four orders-of-magnitude each year. Our results provide insight on how seasonal dynamics affect habitability of near-surface ice and microbial processes in a portion of the glacial biome poised to expand in extent with increasing global temperature and ablation season duration.



## 1 Introduction

Annual melt cycles generate water on the surfaces of glaciers and ice sheets that is transported through a sinuous network of channels to the margin as supraglacial runoff or glacier bed via crevasses and moulins to be discharged subglacially (Smith et al., 2008; Chu, 2014). Large amounts of meltwater are temporarily stored on the surface in
cryoconite holes (Hodson et al., 2008; Edwards and Cameron, 2017), supraglacial lakes (Fitzpatrick et al., 2014; Langley et al., 2016), and near-surface aquifers within the firn or ice (Forster et al., 2014; Hoffman et al., 2014; Karlstrom et al., 2014; Christianson et al., 2015; Cook et al., 2016). Considerable quantities of meltwater have been shown to accumulate in the firn layer, with some temperate glaciers caching an equivalent of 12% to 44% of the total water stored seasonally (Fountain and Walder, 1998), and the capacity of the Greenland Ice Sheet firn
aquifer is estimated between 322 to 1,289 Gt of water (Harper et al., 2012). As such, ice mass loss driven by warming temperatures has motivated numerous efforts to understand how surface meltwater production and hydrology will affect future sea level rise (e.g., Bamber and Aspinall, 2013) and subglacial hydrological processes that influence ice velocity and ice sheet thinning (Zwally et al., 2002; Sole et al., 2011).

    Glacial meltwater production also provides aquatic environments that are suitable for supporting
microbial communities and promoting active biogeochemical cycling on glacier surfaces (Hodson et al., 2008; Edwards and Cameron, 2017). Conditions in certain supraglacial environments even fall within the tolerances of cold-adapted multicellular organisms and invertebrate populations have been documented in a number of cryoconite ecosystems (Zawierucha et al., 2015). Visual discoloration of snow and ice surfaces due to blooms of cyanobacteria and algae are commonly observed (Hopes et al., 2017), and in regions with standing water (e.g.,
cryoconite holes), photosynthetic rates can be sufficient for positive net community productivity (Anesio et al., 2009; Hodson et al., 2013). An important consequence of microbial growth and pigment production is darkening of the ice surface and albedo reduction (Yallop et al., 2012; Stibal et al., 2017; Tedstone et al., 2017); a critical parameter governing surface melting (Gardner and Sharp, 2010).

    In the ablation zone of a glacier, the absorption of shortwave radiation within the ice promotes internal
melting (Müller and Keeler, 1969), producing a highly porous weathering crust in the near-surface that links segments of the supraglacial hydrological system and transports meltwater, microbes, and nutrients (Irvine-Fynn et al., 2012; Hoffman et al., 2014; Karlstrom et al., 2014; Cook et al., 2016). The availability of photosynthetically active radiation (PAR) in the near-surface aquifer may support phototrophic metabolism (Hodson et al., 2013; Irvine-Fynn and Edwards, 2014), allowing the development of microbial communities in the glacial weathering
crust during the melt season. The weathering crust aquifer (WCA) is presumed to be an important component of a glaciers' hydrological and biogeochemical budget (Irvine-Fynn et al., 2012; Cook et al., 2016; Rassner et al.,



2016), but few data have been available to assess microbial activities and biogeochemical transformations in these environments.

This study examined physical, geochemical, and microbiological properties in the weathering crust of the temperate Matanuska Glacier (located in the Chugach Mountains of southcentral Alaska) to gain information on the nature of the near-surface aquatic ecosystem it supports during the ablation season. Comparison of water percolating through the weathering crust to the surface ice melt revealed unique geochemical features and the enrichment of certain microbial taxa in the WCA, supporting that it may possess distinct biogeochemical processes and species within the supraglacial environment. Sufficient PAR fluxes exist with depth in the ice for photosynthetic activity, and the WCA samples contained growing cell populations and phototrophic taxa. We use these biogeochemical data and modeling results to speculate on how microbial processes in the weathering crust are likely affected by seasonal variability of temperature and light as well as discuss the implications of these results to ice masses in a warming climate.

## 2 Methods

### 2.1 Site description

The Matanuska Glacier is a ~48 km valley glacier with an upper accumulation zone at an elevation of ~3500 masl and the terminus is at ~500 masl; its width ranges from ~ 3 km near the equilibrium line to ~5 km along the terminus (Arcone et al., 1995). The measurements and samples analyzed in this study were collected in the vicinity of a research site established on the western lateral margin in the ablation zone of the glacier (61 42' 9.3"N, 147 37' 23.2"W) during June to July of 2014 and 2015, approximately 8 km from the glacier terminus. Science and technical operations were performed on the ice surface and in portable shelters that contained various scientific equipment, including a 61 x 31 cm laminar flow hood (Fungi Perfecti LLC, Item no. E-ALFH1). A meltwater stream ~100 m upglacier from the research camp was sampled to provide a measure of the microbial community structure in supraglacial water environments in proximity to the study site.

### 2.2 Borehole sampling of ice and water

Boreholes were melted into the ice surface using three different approaches. First, water heated using a diesel-fueled water heater (Pressure-Pro Inc.) was circulated through a tapered aluminum melt head (10 or 30 cm in diameter; 30 cm in length) that was attached to a standard garden hose and manually guided while descending through the ice. Secondly, an electrically-heated copper melt head of 10 cm was used to create boreholes of up to



8.5 m depth. The electrothermal heater was powered with a portable 2.5 kW generator, with the primary front heater reaching up to 2 kW power output and enabling ice penetration rates of 1-2 m per hour. Finally, 25 to 30 cm boreholes were generated with a laser-powered ice-penetrating cryobot that heated the water and used on-vehicle hot water jets to melt the ice and control its direction and rate of descent (Stone et al., 2014, 2018). In June

2014, four −30 cm diameter boreholes (BH1-3 and BH-5; 4 to 30 m in depth) and one −10 cm diameter borehole (BH4; 4 m in depth) were used to sample ice and water in the WCA (Supplementary Table 1). The same location on the glacier was revisited during June to July 2015, and two-30 cm diameter boreholes (BH6 and BH7; 1.4-15.4 m in depth) and three-10 cm diameter boreholes (BH8-10; 4 to 5 m in depth) were sampled (Supplementary Table 1).

In the 30 cm diameter boreholes, particulates in large volumes of meltwater (up to 404 L) were size fractionated (i.e., particulate fractions >3 μm, <3 to >0.8 μm, and <0.8 to >0.2 μm) and concentrated in situ on 142 mm Supor PES membranes (Pall) using a McLane Large Volume Water Transfer System (WTS-LV; Christner et al., 2014). Water in the boreholes was also sampled by lowering clean silicone tubing into the borehole and using a peristaltic pump to deliver the water to the surface. To determine if the ice was porous and active

water flow was occurring through the weathering crust, water was drained from several of the boreholes and monitored to assure that melt water did not enter directly from the surface. Two of the drained boreholes refilled with water that percolated laterally from the WCA (BH4 and BH10; Supplementary Table 1); this material was retrieved at the surface using a peristaltic pump. For chemical analysis, WCA water samples were collected by lowering clean acid-washed HDPE bottles into the borehole using nylon paracord.

Meltwater samples from discrete ice depth intervals were obtained using a customized water sampler and filtration system that was integrated as a scientific payload on the cryobot (Clark et al., 2017; Stone et al., 2018). Operation of the cryobot requires a water-filled cavity; therefore, the 30 cm aluminum melt head was used to create a ~1 m pilot hole, the meltwater generated was replaced with an equal volume from a supraglacial stream, and the water was continually circulated through a 0.2 μm filter for ~1 h to reduce the number of microorganisms

in the priming water. Stainless steel SwageLok fittings and tubing from the melt head were connected to a peristaltic pump, which was used to pump water into sterile 120 mL capacity sampling bags (Labtainer™ BioProcess Container, cat. No. SH30658.12; 10 total per mission) or through 90 mm Supor PES filters with a 0.2 μm pore size (5 total per mission). The sampling bags had duel ports: one for filling through a sanitary check valve and one for overflow, which was controlled by a sanitary pressure relief valve. The filters were housed

within a Savillex filter holder made of PFA fluoropolymer and sealed with Ultem clamps. During water or filtration sampling, the cryobot was operated in "passive" mode (i.e., the hot water jets were not used and heat



transfer was directed only to the melt head of the vehicle; Clark et al., 2017), which melted the ice at a rate of ~1 L min$^{-1}$. To prevent inadvertent collection of bulk borehole water, the sampling pumps were operated at a rate of 56 mL min$^{-1}$ (i.e., ~6% of the rate of melt water generation). Based on the rate of descent and collection, each water sample of 120 mL represented the melt generated from a depth horizon of approximately 1.6 cm in the ice.

## 2.3 Conductivity, pH, and chemical composition of the water samples

Electrical conductivity and pH of the water samples were measured using a multi-parameter PCSTest probe (Oakton Instruments, Vernon Hills, IL).

The concentration of major ions and trace elements in the water samples were analyzed using inductively coupled plasma mass spectrometry with a reaction cell for cations (ICP RC-MS, Agilent 7500c) and ion chromatograph (Dionex ICS 5000+) for anions. For ICP RC-MS analysis, samples were acidified to 1% HNO$_3$ v/v and analyzed for 28 elements. Elements prone to interferences (e.g. Ca$^{2+}$ and As$^{3+}$) were analyzed using H$_2$ and He as the reaction gas. Quantification was performed using seven external calibration standards ranging from 0.1 to 100 ppb. Drift correction was achieved by online addition of 10 ppb of a four element internal standard mix (Li(7), Y, Ce, and Bi). An IonPac AS15 2x250 mm column was used for anion separation using 38 mM KOH as eluent and ASRS 300 zero reagent suppressor. The sample injection volume was 10 µL and quantification was performed using five external calibration standards ranging from 0 ppm to 10 ppm. Calibration verification standards and blanks were run every 10$^{th}$ analysis for anions and cations. The NIST SRM 1643d was used to verify cation calibration and a secondary anion standard (Anion II Std Dionex) was used to check anion calibration. Samples that exceeded the calibration by 120% were diluted and reanalyzed. Nitrite was not analyzed due to overlap with bicarbonate in the chromatogram. The mass balance deficit between anions and cations was used to determine bicarbonate concentrations.

## 2.4 Ice temperature and meltwater content

Vertical gradients of temperature in the ice surface were measured over two months during the summer of 2015 by deploying a sensor string into a 45 m-deep borehole with thermistor sensors placed at every 10 m along the string and collecting data at hourly intervals. The thermistors were calibrated under controlled conditions and had an accuracy of ca. 0.1 ᵒC.

A one-dimensional finite-difference model utilizing the explicit Euler central difference method was used to calculate ice temperatures within 100 m of the surface at 0.25 m vertical intervals and with hourly time step. Since Matanuska Glacier is temperate, the lower boundary condition was set to the pressure-melting point while



the surface condition was driven using the 2 m-air temperature and the surface downwelling shortwave flux in air. Because these quantities are not available from nearby weather stations, the European Centre for Medium-Range Weather Forecasts ERA Interim reanalysis data for the 0.75 by 0.75 degree cell containing this region of the Matanuska Glacier for the period between the beginning of 2014 and the end of 2016 was used (Fig. 1B; Dee et

al., 2011). Air temperature is taken to be equal to the surface temperature at the top of the model domain when the former is negative but is set to the melting point of ice when air temperature is positive. When the latter is the case, the surface melt rate was calculated using the Positive-Degree-Day parametrization (van Beusekom et al., 2010) and a melt factor of 5 mm per degree Celsius per day. The surface melt rate, which typically reaches several cm per day in summer (Ensminger et al., 1999; Reynolds, 2005; Mankoff and Russo, 2013), was treated

throughout the 100 m-thick model domain as spatially uniform upward advection. Surface solar radiation downwards from the ERA-interim dataset was used to scale internal heating in the upper, optically translucent part of the glacier following the exponential Lambert Law of light attenuation with e-folding length scale of 0.825 m as constrained by the PAR data (Fig. 2A). Heat conduction was treated as a diffusive process using a diffusion coefficient of $1.2 \times 10^{-6}$ $m^2$ $s^{-1}$. The model neglects the impact of seasonal snowpack on either the temperature or

energy balance of the simulated glacier surface. The entire model domain was initialized as temperate ice and then the calculation cycled through repeats of the 3-year surface forcings until the drift of calculated ice temperatures reached <0.01 ºC over a 3-year cycle.

### 2.5 Determination of the PAR fluence rate

PAR (400-700 nm) was measured in boreholes using a LI-COR LI-193 Spherical Underwater Quantum Sensor

(LI-COR, Inc) and 10 m underwater cable. The stated sensitivity of the LI-193 is typically 7 μA per 1,000 μmol $s^{-1}$ $m^{-2}$ in water. PAR was also measured at the ice surface with a LI-COR quantum sensor mounted on a small board using a leveling device. Both sensors were driven and logged by a LI-COR LI-1400 Data Logger and were calibrated prior to the first field season.

      Two types of measurements were made in water-filled boreholes: vertical profiling within a short period

of time and overnight measurement at a fixed depth. For profiling, the sensor was deployed in a 10-cm diameter borehole and measurements were taken from 0 to 8.5 m at 0.5 m increments. To mitigate the "borehole effect" caused when measuring transmitted light in a liquid-filled borehole, the sensor was placed 1 m below a 10-15 cm opaque packer (Muni-Ball®, Cherne Industries Inc.) that was inflated at discrete depths to create a light-tight seal. To avoid freezing the packer in the ice during overnight deployments, the sensor was deployed at 3.5 m in a 30

cm diameter borehole and light was blocked from entering with an opaque tarp. Due to the larger distance between



blocked light and the sensor, the diel measurement may have been subject to a larger borehole effect than the profile measurement.

The PAR attenuation coefficient ($K_{PAR}$) was derived as described by Hodson et al. (2013) based on $PAR(Z) = PAR_0 e^{-KZ}$, where Z is the ice depth from the surface in meters and $PAR_0$ is the surface PAR flux in

$\mu mol\ m^{-2}\ s^{-1}$.

## 2.6 Epifluorescence microscopy

The water samples collected for epifluorescence microscopic analysis were immediately fixed by addition of sodium borate-buffered formalin (5% final concentration), stored at ~4°C, and analyzed within 1 month of collection. Cells were concentrated by filtering 10 to 40 mL of sample onto 0.22 µm black Isopore filters

(Millipore, cat. no. GTBP02500) and stained for 15 minutes in the dark with SYBR$^{TM}$ Gold (Molecular Probes Inc., cat. no. S-11494) buffered in Tris-borate-EDTA (pH 8.3). The staining solution was pulled through the filter by applying vacuum, the filter was mounted on a microscope slide, and a drop of antifade (0.1% phenylenediamine in a 1:1 ratio of PBS and glycerol) was added to the middle of the filter before applying a coverslip. For each sample, the number of DNA-containing cells in sixty fields of view (FOV) was enumerated using an Olympus

BX51 epifluorescence microscope. The cell concentration was estimated by calculating the average number of cells per FOV, multiplying by the number of FOVs per filter, and dividing by the volume of sample filtered.

## 2.7 Determination of ATP biomass

Triplicate measurements of ATP were performed on each sample in the field by concentrating the cells in 25 mL of water on a 0.22 µm Millex-GS Syringe Filter (Millipore, cat. no. SLG033SS), followed by passing 0.2 mL of

Extractant B/S from the ATP Biomass Kit HS (Biothema Inc., cat. no. 266-112) and collecting the filtrate. For each procedural replicate, one-quarter of the filtrate (50 µL) was added to 400 µL of ATP Reagent HS and luminescence was measured using a 20/20n luminometer (Turner Biosystems, cat. no. E5331). Subsequently, 1 pmol of an ATP standard was added to each sample and luminescence was measured, allowing the relative luminescence units (RLUs) for each measurement to be corrected for possible inhibition of the luciferase enzyme

and/or changes of ambient temperature during the course of measurement. The quantity of ATP was determined by generating standard curves (six 10-fold dilutions of 100 nM to 1 pM ATP) that allowed relating RLUs to the ATP concentration.



**2.8 Detection of respiring cells via reduction of a tetrozolium salt**

An assay with the tetrazolium salt XTT [2,3-Bis(2-methoxy-4-nitro-5- sulfophenyl)-2H-tetrazolium-5- carbox-anilide] was performed in the field to assess cellular respiration activity. The water sample (50 L) was filtered through a 142 mm, 0.22 µm SUPOR filter (Pall) and the retentate was suspended in a volume of filtrate that

concentrated material > 0.2 µm in diameter approximately 1500-fold. Aliquots of this suspension (1 mL) were amended with the XTT reagent (XTT Cell Proliferation Assay Kit, Trevigen Cat. No. 4891-025-K) to a final concentration of 330 µM; the electron-coupling agent (phenazine methosulfate) was not included in the assay (see Roslev and King, 1993).

Individual samples in 1.5 mL microcentrifuge tubes were placed in 50 mL conical tubes that were either

transparent or covered with black electrical tape (i.e., dark incubation), affixed to a nylon tether, and incubated in the borehole at a depth of 1.5 m below the surface. A packer (Muni-Ball®) was inflated 1 m above the samples to prevent direct light transmittance through the water in the borehole. At designated time points, formazan production was determined in triplicate samples by measuring absorption at 470 nm with an Ocean Optics USB4000-UV-VIS miniature spectrometer.

**2.9 Dissolved oxygen concentration and oxygen uptake rates**

The concentration of dissolved oxygen was measured using Winkler titration and the Dissolved Oxygen Test Kit (Hanna Instruments, cat. no. HI3810). To determine oxygen uptake rates, samples of the water were dispensed into serum vials (Wheaton, cat. no. W223743) and sealed with butyl rubber stoppers while ensuring that no air bubbles were present in the headspace. During transport from the field site and shipment to the laboratory at

Louisiana State University, the samples were kept chilled on blue ice and stored in the dark. Two weeks after sample collection, the serum vials were transferred to an environmental chamber, temperature probes were placed directly above the vials, and the samples were incubated at 5ºC. All samples were incubated under a continuous illumination of ~270 µmol photons m$^2$ s$^{-1}$, with half of the vials covered with aluminum foil for dark incubation. Over the course of 117 days, triplicate measurements of dissolved oxygen concentration and cell abundance were

made on the samples. Rates of oxygen uptake and cell growth at 5ºC were adjusted to the in situ temperature of 0ºC using the Arrhenius equation and an energy of activation of 15,000 cal mol$^{-1}$ (Priscu, 2013).





## 2.10 DNA extraction, amplification of 16S/18S rRNA gene sequences, and phylogenetic analysis

The filters obtained for nucleic acid extraction were quartered, placed in cryotubes containing a solution of 40 mM EDTA pH 8.0 and 50 mM Tris pH 8.3, and stored chilled in an insulated cooler on the glacier for approximately 1 week. Immediately after return from the field, sucrose was added to the storage buffer (final

concentration of 0.73 M), the cryotubes were frozen, and shipped overnight to Louisiana State University on blue ice. The samples were stored in a laboratory freezer at -80ºC until analyzed.

To extract DNA from the filters, a sterile scalpel was used to cut the filter into small pieces to enhance yield during bead beating. Sterile forceps were used to transfer the filter pieces to bead beating tubes. The storage buffer remaining in the cryotubes was centrifuged at 16,000 x g for 10 min to pellet any particulates remaining in

the storage buffer, and this material was added with the filter to the bead beating tubes. DNA was extracted from the samples using the Power Water DNA Isolation Kit (MO BIO Laboratories, Inc.) with the following modifications of the manufacturer's instructions. First, some of the extracted filters were separated into two bead beating tubes for the initial steps of the protocol, and subsequently, the supernatants were combined on a single spin filter. Second, the samples were incubated at 65° C for 10 min after adding the lysis buffer followed by bead

beating in a Mini-Beadbeater-16 (Bio Spec Products Inc.) at 3450 oscillations min$^{-1}$ for 4 to 5 min. Lastly, the filter was washed with an additional volume of Solution PW1, vortexed, centrifuged, and the supernatants were pooled. The genomic DNA was suspended in 10 mM Tris buffer, quantified by absorbance at A$_{260}$ using a Nanodrop (Thermo Fisher Scientific) or with a Qubit fluorometer (dsDNA HS Assay Kit, Thermo Fisher Scientific), and stored at -20°C. Three blank filters were included in the DNA extraction protocol to serve as

procedural controls for the molecular analysis.

A portion of the 16S rRNA gene (V4 region) was amplified from the extracted genomic DNA samples using the oligonucleotide primers 515F and 806R (Caporaso et al., 2012). Each 25 or 50 µL PCR reaction contained 0.1 to 6 ng of genomic DNA, 3 mM MgCl$_2$, 1X Gold buffer, 0.2 mM dNTP's, 0.2 µM of each primer, and 2.5 U of AmpliTaq Gold DNA polymerase LD (Invitrogen). The amplification conditions included an

incubation at 95ºC for 8 to 10 min followed by 35 to 45 cycles of denaturation at 94ºC for 30 s, annealing at 50ºC for 30 s, and extension at 72ºC for 45 s. A final extension was performed at 72ºC for 5 min. The amplicons were evaluated by electrophoresis through a 1% agarose gel, purified using the MoBio UltraClean PCR Clean-Up Kit, quantified with Qubit fluorometry, and 240 ng of each amplicon was pooled. Paired-end sequencing was performed using the Illumina MiSeq platform (RTSF Genomics Core at Michigan State University and ICBR

NextGen DNA Sequencing Core at University of Florida).



Barcodes and adapters were removed using Trimmomatic (v0.36; Bolger et al., 2014) and the amplicon sequences were processed using mothur (Schloss et al., 2009). Paired end sequence reads of the V4 region were assembled into contigs and quality filtered using the following default parameters: maximum sequence length of 275, minimum quality score of 25, and maximum ambiguous bases of 0. Sequences were aligned with the mothur

compatible SILVA database (v128). Chimeric sequences were removed using the VSEARCH algorithm within mothur. The mothur-formatted Ribosomal Database Project (RDP) training set (v16) was used to assign all sequences with ≥97% similarity to an operational taxonomic unit (OTU). Singletons and OTUs in the procedural controls were removed before data normalization. To assess community diversity and richness of the samples, the OTU abundance data were normalized (9,151 sequences); Good's estimator of coverage and Simpson's diversity

index were calculated in mothur. Abundant OTUs were defined as having a relative abundance of ≥ 0.1% of the total sequences. Non-metric multidimensional scaling (NMDS) plots were calculated within mothur and visualized in R (v.3.4.1; R Core Team, 2017).

A portion of the 18S rRNA gene (V4 and V5 regions) was amplified using genomic DNA isolated from sample BH10d and the universal eukaryal primers F566 and R1200 (Hadziavdic et al., 2014). PCR reactions of

25 µL contained 2.8 ng of genomic DNA, 1X Taq buffer A, 0.5 mM dNTP's, 0.5 µM of each primer, and 1.25 U of Taq polymerase (Fisher BioScience). The amplification protocol started with a 3 min incubation at 95°C, followed by 35 cycles of denaturation at 95°C for 30 s, annealing at 60°C for 45 s, and extension at 72°C for 1 min. A final extension was done at 72°C for 10 min. Gel electrophoresis through a 1% agarose gel was used to evaluate amplicon size and clone libraries were constructed with the TOPO TA Cloning Kit for Sequencing

(Invitrogen) according to the manufacturer's instructions. Individual clones (25) were cultured in Miller LB media containing 50 µg mL$^{-1}$ of ampicillin and plasmid DNA was purified using the PureLink Quick Plasmid Miniprep Kit (Invitrogen). Plasmid DNA was quantified with a Nanodrop 2000c (Thermo Scientific) and 50-150 ng of each was used to bi-directionally sequence each insert by Sanger sequencing (Eton Bioscience, Inc.).

Phylogenetic tree construction was conducted by aligning the query sequences based on secondary

structure to their nearest neighbors using SINA Alignment Services (Pruesse et al., 2012). The sequence alignment was trimmed using MEGA7 and a phylogenetic tree was created with the maximum likelihood method (Kumar et al., 2016). Bootstrap values were calculated in MEGA7 based on Felsenstein's method and 1000 replicates.

**2.11 Statistical analysis**

The assumption of normality was evaluated using the Shapiro-Wilk test. The strength of linear and montonic

correlations between geochemical and microbiological variables was assessed by evaluating Pearson's and



Spearman's correlation coefficients, respectively. Student's t-distribution test was applied to determine the probability of significant differences between variables ($\alpha \leq 0.05$). Statistical differences between the microbial communities were evaluated using analysis of molecular variance (AMOVA) in mothur (Schloss et al., 2009).

## 3 Results

### 3.1 Evidence for a porous weathering crust and near-surface aquifer system

During the first eleven days of field work in June 2014, there was no evidence for porous flow of water into boreholes that had been drained of meltwater. However, borehole 4 (BH4; hereafter, all boreholes are similarly abbreviated as designated in Supplementary Table 1) gradually filled during the final two days of observation and samples of the water were collected on 20 June 2014 (BH4b). From late June to early July 2015, the same location on the glacier was revisited and similar observations were made. Surface melt water subsequently infiltrated two of the boreholes (BH8 and BH9) and only BH10 remained isolated from direct surface input; therefore, it was used as the source for the WCA water samples collected in 2015. BH10 was melted on 25 June 2015, and based on a slow initial rate of refill combined with uncertainly about the contribution of runoff from two days of rainfall, the water was drained after approximately one week and discarded. Opportunely, subsequent refill of BH10 on three consecutive days (5, 6, and 7 July 2015; samples BH10b, BH10c, and BH10d, respectively; Supplementary Table 1) provided a total of ~110 L of water from the WCA for analyses.

### 3.2 Physical properties of the near-surface ice

A one-dimensional numerical model of heat diffusion, advection, and production was used to create a prediction of temperature (Fig. 1A) and subsurface melt distribution in the ice based on internal heating (Fig. 1B). Modeled temperatures are dominated by the effects of downward propagation of the winter cold wave to depths of ~10 m (Fig. 1A). Somewhat more subtle is the development of an isothermal, temperate summer layer in the top ~2 m of the glacier due to internal heating caused by penetration of solar radiation. Once at the melting point, this layer experiences internal melting resulting in water contents up to 200 L m$^{-2}$ or ~10% by volume (Fig. 1B). The onset of freezing air temperatures in winter quickly eliminates the water-bearing layer in the model.

Comparison of the modeled and observed temperatures (Fig. 1C) shows limited agreement. The most striking differences are observed at the lowermost sensor at 45 m, which should be constantly at the pressure melting point according to the model. Sensors at 35 m and 25 m were more uniformly temperate through time, except for slight cooling in the second half of August 2015, where the 25 m sensor cooled below freezing sooner



than the sensor at 35 m. The behavior of the shallower three sensors are consistent with them being frozen within ice, but the sensor at 45 m was likely within, or near, an air-filled cavity, at least after ca. 25 July 2015 when its recording became highly variable. Neither the magnitudes nor the high variability of temperatures can be explained if the sensor was only interacting thermally with solid ice at 45 m depth. Surface observations in this

ice marginal area confirmed the presence of many crevasses and crevasse traces, as well as moulins. Hence, the disagreement between modeled and observed temperatures is attributed to the fact that the former does not include such heterogeneities, which play an important role in transporting water and heat within the ice column. The model also makes no provision for meltwater percolation and storage in intercrystalline veins. Where fast drainage pathways exist (i.e, crevasses and moulins), the meltwater table can drop well below the surface and permit air

access and rapid temperature changes, as indicated by the 45 m sensor (Fig. 1C).

     To examine the optical properties of the near-surface ice and its potential to support photosynthetic activity, the diel flux of incident PAR (400-700 nm) and PAR attenuation with ice depth were determined (Fig. 2). Based on mid-afternoon borehole measurements on 16 June 2014, PAR intensity decreased exponentially with ice depth (Fig. 2A) and is described well by the equation: Fraction of PAR flux (relative to surface) $= e^{(Z + 0.3621 / -}$

$^{0.7017)}$, $r^2 = 0.9926$; where $Z$ is the ice depth from the surface in meters. The PAR attenuation coefficient ($K_{PAR}$), calculated according to Hodson et al. (2013), is 1.5 m$^{-1}$. Based on PAR data for benthic mats (1 µmol m$^{-2}$ s$^{-1}$; Hawes and Schwarz, 2001) and the water column of highly shaded Antarctic lake ecosystems (0.43 and 0.07 µmol m$^{-2}$ s$^{-1}$; Priscu et al., 1988), midday PAR fluxes could support photosynthesis to maximum depths of 5 to 7 m (Fig. 2B).

### 3.3 Chemical composition of the near-surface ice and WCA

A novel ice penetrating cryobot (Stone et al., 2014, 2018) with a water sampling system (Clark et al., 2017) was used to obtain melt water from two boreholes (BH6 and BH7; Supplementary Table 1) located within ~1.5 m of each other, and 24 samples were collected from discrete depths between 1.4 and 8.3 m and filtered for chemical analysis. These data (Supplementary Table 2) indicated Ca$^{2+}$ was the dominant cation (average of $56 \pm 27$ ppb; $\pm$

standard deviation) and that it strongly positively correlated ($r_s > .62$, n = 24, $p < 0.001$) to subordinate K$^+$, Na$^+$, Mg$^{2+}$, Al$^{3+}$, and Fe$_{(aq)}$ concentrations. Sulfate was the most abundant anion ($136 \pm 28$ ppb) followed by HCO$_3^-$ ($75 \pm 78$ ppb), the latter of which showed large variation with depth (Fig. 3A; Supplementary Table 2). Nitrate and Cl$^-$ exceeded the analytical limit of detection in approximately one-quarter of the samples in the profile, averaging $81 \pm 21$ and $172 \pm 38$ ppb, respectively.



Water that percolated into BH10 via porous flow during July 2015 had a pH of 6.2 and 6.8, and conductivity of 12 and 4 $\mu S$ cm$^{-1}$ (samples BH10b and BH10d, respectively), which was a similar pH but higher conductivity than that observed in supraglacial waters (~1 $\mu S$ cm$^{-1}$). Chemical analysis of WCA sample BH10b (Supplementary Table 2) indicated that in contrast to data obtained from ice melt, Na$^+$ was the dominant cation

and Cl$^-$ was the dominant anion. Comparing odds ratios for each analyte indicated that six compounds were substantially enriched (4- to 170-fold; Na$^+$, K$^+$, F$^-$, Al$^{3+}$, Fe$_{(aq)}$, and Br$^-$) and five were depleted (3- to 17-fold; SO$_4^{2-}$, NO$_3^-$, Cr$^{3+}$, Pb$^{2+}$, and Co$^{2+}$) in the WCA sample relative to average concentrations in ice melt from 1.4 to 8.3 m (Fig. 3B). The largest differences observed between the ice and WCA chemistry were with F$^-$ (170- fold enriched in the WCA), Pb$^{2+}$, and Co$^{2+}$, (16- and 17- fold depleted in the WCA, respectively).

**3.4 Microbial cells and biomass in the near-surface glacial ice and WCA**

The water sampling system of the cryobot was used to collect 45 discrete samples at depths between 1.4 and 11.4 m to estimate the abundance of microbial cells and cellular ATP (Fig. 4A). The Spearman rho value ($r_s$ = .38, n = 45) implied a weak positive correlation between the cell and ATP concentration data, but the correlation was not statistically significant ($p$ = 0.066). Nevertheless, the cell and ATP data profile showed similar trends, with the

highest concentrations of each at depths of 1.4 to 1.6 m (816 ± 270 cells mL$^{-1}$ and 155 ± 83 amol L$^{-1}$, n = 5) and lowest values observed at depths > 7.2 m. At depths below 2 m, there were three horizons (4.0 to 4.1, 6.3 to 6.4, and 8.3 m) where the cell and ATP concentration were higher relative to samples from adjacent depths (Fig. 4A). The cell and ATP data strongly positively correlated with Co$^{2+}$ ($r_s$ >.60, n = 24, $p$ < 0.002) and moderately positively correlated with HCO$_3^-$ ($r_s$ > .47, n = 24, $p$ < 0.019); the ATP concentration also moderately positively

correlated with Al$^{3+}$ and K$^+$ ($r_s$ > .57, n = 24, $p$ < 0.004; Fig. 3A).

The cell concentration in the water generated by melting BH10 (i.e., BH10a) was similar to average values from depths of 1.4 to 11.3 m (Fig. 4C) and 875-fold lower than that observed in supraglacial water (1.92 ± 0.465 x 10$^5$ cells mL$^{-1}$). Relative to the 2$^{nd}$ refill event collected on 5 July 2015 (BH10b; 2,760 ± 488 cells mL$^{-1}$), subsequent sampling on consecutive days (6 and 7 July 2015; samples BH10c and BH10d, respectively) revealed

that cell concentrations in the 3$^{rd}$ and 4$^{th}$ refills were 78% and 93% lower, respectively (Fig. 4C). The ATP concentration in water sampled from the 3$^{rd}$ refill (BH10c) was 32 ± 3.4 amol L$^{-1}$ and slightly lower than the average for depths of 1.4 to 11.3 m (54 ± 50 amol L$^{-1}$; Fig. 4A).



### 3.5 Microbial respiration and oxygen consumption

The respiration potential of microorganisms in the WCA was evaluated in June 2014 by measuring the reduction of XTT [2,3-Bis(2-methoxy-4-nitro-5- sulfophenyl)-2H-tetrazolium-5- carbox-anilide] during time-course experiments that were incubated at a depth of 1.5 m in a packered borehole. Sample for the rate measurement was

obtained from water in BH5, which was melted one day prior to the experiment. To investigate if the respiration rate was affected by PAR availability at 1.5 m (Fig. 2), identical preparations of the samples were incubated in situ in the light and dark. Under both conditions, the production of formazan at the in situ temperature was linear over 5.5 h (Fig. 5A), and the slope of the data from the light ($b_{light}$ = 0.0691) was slightly higher than that in the dark ($b_{dark}$ = 0.0610), but formazan production was not statistically different between the treatments (Student's t-

distribution, p = 0.46, n = 29).

During July 2015, water collected from the $3^{rd}$ refill event in BH10 (sample BH10c) was found to be under-saturated with respect to oxygen (11.6 ± 0.3 mg $L^{-1}$, 88% of air-saturated water). To estimate the oxygen uptake rate in the WCA, the BH10c samples were sealed in serum bottles with no headspace and incubated ex situ at 5°C in the dark and light (~270 μmol photons $m^2$ $s^{-1}$) for 117 and 97 days, respectively. The rate of oxygen

consumption during these experiments (Fig. 5B) was modeled as a first order decay process and revealed that samples incubated in the light had a larger reaction rate coefficient (k = 0.0167 $d^{-1}$, $R^2$= 0.644) than those in the dark (k = 0.00553 $d^{-1}$, $R^2$= 0.676). Data from the first 32 days of incubation were used to calculate the initial rate of consumption in the light and dark (152 and 70 μg $O_2$ $L^{-1}$ $d^{-1}$, respectively), and the linear regression models for these conditions had significantly different slopes (p = 0.010, n = 24). Correction of the rate data to the in situ

temperature of 0°C (Fig. 1A) provided oxygen consumption estimates of 42 and 92 μg $O^2$ $L^{-1}$ $d^{-1}$ in the dark and light, respectively.

The WCA sample used for the ex situ incubations (BH10c) had an initial cell density of 620 ± 130 cells $mL^{-1}$ (Fig. 4C). To confirm that oxygen consumption was associated with microbial growth, the cell concentration was determined approximately midway (60 and 46 d for dark and light incubations, respectively) and at the

termination of the experiment. Over the time course shown in Fig. 5B, cell density increased by three orders of magnitude and was 6.3 ± 0.58 x $10^5$ and 5.9 ± 0.45 x $10^5$ cells $mL^{-1}$ for samples incubated in the dark and light, respectively. Assuming exponential growth during the first half of the experiment, the mean doubling time of the populations at 0°C was estimated at 14 and 11 days in the dark and light, respectively.





### 3.6 Composition of microbial assemblages in the near-surface ice and WCA ecosystem

A total yield of ≤ 200 pg (i.e., limit of detection for the DNA quantification method) to 200 ng of genomic DNA was obtained from the filter samples extracted, and a portion of this was used to amplify the V4 region of the 16S rRNA gene. Amplicons obtained from 31 samples (Supplementary Table 3) and four procedural controls (three

DNA extraction blanks and a PCR control, all amplified using 45 cycles) were sequenced, producing 14,349,197 paired-end contigs with an average read length of 253 bp. After quality filtering and removing chimeras and singletons, 10,024,390 sequences remained that classified as 12,970 operational taxonomic units (OTUs). OTUs that co-occurred in the procedural control data at abundances > 0.05% of total sequences (279 OTUs) were designated as experimental contaminants and removed from the dataset, leaving 7,754,705 unique reads for

evaluation. Each sample was normalized to 9,151 sequences, reducing the dataset to 283,681 sequences that classified as 3,381 OTUs. Abundant OTUs were defined as having relative abundances ≥0.1% of total sequences and represented between 65% and 99% of the OTUs in each sample. Based on Good's coverage estimates for the normalized sequences (> 0.96; Supplementary Table 3), the data are sufficient to describe the most abundant taxa in the samples.

15       The abundant OTUs in the samples classify within 9 bacterial phyla: Proteobacteria were the dominant phylum, followed by Cyanobacteria, Bacteroidetes, Actinobacteria, Armatimonadetes, Firmicutes, Acidobacteria, Deinococcus, and candidate phylum WPS-2 (8 bacterial OTUs could not be classified at the phylum level). Based on the NMDS analysis (Fig. 6A), the samples cluster into three highly significantly different groups (AMOVA; p ≤ 0.001, n = 31) that can be generally characterized as: i) 2014 samples from boreholes of 10 to 30 m depth (top

right of plot), ii) 2015 samples collected at discrete depths between 2 to 15 m (top left), and iii) samples heavily influenced by surface and in-ice melting (i.e., stream and WCA water; bulk ice melt from boreholes ≤ 4 m; bottom of Fig. 6A). The vast majority of sequences in the englacial 2014 samples from borehole depths of 10 to 30 m were proteobacterial (94% of the sequences), with taxa in the Actinobacteria (3%), Bacteroidetes (1%), Cyanobacteria (1%), and 6 other phyla comprising the remainder (Supplementary Fig. 1). Samples from the

deepest 2015 boreholes had similar phyla distributions; however, the 2014 samples were dominated by Gammaproteobacteria whereas betaproteobacterial OTUs were more abundant in the 2015 data. This contrasted to the assemblages in the near-surface samples, which were comprised of OTUs affiliated with the phyla Proteobacteria (39%), Cyanobacteria (22%), Bacteroidetes (18%), Actinobacteria (9%), Armatimonadetes (2%), Acidobacteria (1%), Deinococcus/Thermus (1%), and candidate phylum WPS-2 (<0.5%; Supplementary Fig. 1).

30       To explore the provenance of species in the glaciers' near-surface and ascertain if the observed distributions were consistent with the release of microbes from englacial ice during melting, an odds ratio for each





OTU was calculated by comparing abundance with data from the deepest boreholes (Fig. 6B). For the 2014 samples, OTU abundances in samples BH3, BH4a, and BH5 were compared to data from BH2 (n=8; Supplementary Table 3). Similarly, samples from shallow 2015 boreholes (BH9 and BH10a) were compared with data from BH6, BH7, and BH8 (n=7). Based on this approach, 23 OTUs in seven bacterial phlya were identified

that had > 5-fold higher abundances in the glaciers' near-surface (Fig. 6B) and represented approximately one-third of the sequences in these samples. Most of these OTUs classify as Alphaproteobacteria, Bacteroidetes, and Cyanobacteria, with the nearest neighbors for three of the five cyanobacterial OTUs being the 16S rRNA sequences of plastids most closely related to those possessed by chrysophyte and streptophyte algal species (Supplementary Fig. 2). Several of the plastid 16S rRNA sequences were the most abundant OTUs in the near-

surface samples (OTU8 and OTU9 together make up 16% of the 2014 and 11% of the 2015 data). DNA from BH10d was used to amplify and clone a portion of the 18S rRNA gene for phylogenetic analysis. Sequencing of 25 clones identified 3 OTUs related to green and golden algae (Fig. 6C) as well as 4 OTUs related to fungal, ciliate, and rotifer species (Supplementary Table 4). The phylogenetic relationships of the 18S rRNA gene sequences related to members of the classes Zygnematophyceae and Chrysophyceae (Fig. 6C) was consistent with

the composition of algal taxa inferred based on the plastid 16S rRNA gene data (Supplementary Fig. 2).

**4 Discussion**

Near-surface internal melting of glacial ice provides liquid water for microbial metabolism and promotes meltwater percolation that can transport solutes, gases, organic matter, and cells both vertically and horizontally through a highly permeable ice layer (Irvine-Fynn et al., 2012; Cook et al., 2016). To investigate the parameters

influencing the transformation of polycrystalline ice into a perched aquifer and biogeochemically-active habitat during the ablation season, a one-dimensional heat conduction-production model was used to examine the effect of temperature and solar radiation on water availability in the ice. Elevated air temperatures during summer combined with heat from the dissipation of shortwave radiation produce a ~2 m solar weathering crust that contained a meltwater fraction as high as 10% of the ice by volume (Fig. 1). The depth of meltwater production

inferred by the model is consistent with observations from High Arctic glaciers (Irvine-Fynn and Edwards, 2014) but exceeds that for Antarctic dry valley glaciers by more than 10-fold (5-15 cm; Hoffman et al., 2014). Based on surface ice densities as low as 0.5 g cm$^{-3}$ (Müller and Keeler, 1969), total water content in the weathering crust of some glaciers may approach 50%.



Irvine-Fynn and Edwards (2014) speculated that environmental conditions in the WCA facilitate biogeochemical processing by providing microbes with access to light, effectively extending a glacier's "photic zone" to depths beneath the surface. The PAR attenuation coefficient derived for surface ice of the Matanuska Glacier during summer 2014 ($K_{PAR} = 1.5$ m$^{-1}$) indicates higher light propagation with depth compared to wet or dry snow ($K_{PAR} = 7.5$ to 20 m$^{-1}$) and is similar to values reported for lake, sea, and blue ice (Hodson et al., 2013 and references within). Based on the photosynthetic properties of highly shaded polar aquatic ecosystems (Priscu et al., 1988; Hawes and Schwarz, 2001), sufficient PAR may penetrate to ice depths of 5 to 7 m during daily maximums to support cyanobacterial and algal photosynthesis (Fig. 2B). However, since the solar radiation flux is incapable of producing appreciable meltwater at depths > 2 m (Fig. 1A), photosynthetic and microbial activity are likely localized to the near-surface region. The upper 2 m of ice experiences at least 3% of the surface PAR flux (Fig. 2A) and may hold up to 200 L of liquid water per square meter of glacier surface during the peak of the melt season (Fig. 1B). Remarkably, if the WCA community possesses species capable of acclimating to highly shaded conditions, such as those found in ice-covered Antarctic lakes (0.07 µmol m$^{-2}$ s$^{-1}$; Priscu et al., 1988), then the diel pattern observed during civil twilight implies the PAR flux continually exceeded the threshold necessary to support photosynthetic activity (Fig. 2B).

The oxygen uptake rate for WCA samples incubated in the light (92 µg O$_2$ L$^{-1}$ d$^{-1}$; Fig. 5B) was more than twice values obtained in the dark (42 µg O$_2$ L$^{-1}$ d$^{-1}$) and ~5-fold lower than estimates in supraglacial melt ponds in proximity to the study site (440 to 600 µg O$_2$ L$^{-1}$ d$^{-1}$ under ambient light; data not shown). Similar measurements in most aquatic ecosystems typically observe higher dissolved oxygen concentrations when samples are incubated in the light due to stimulation of oxygenic photosynthetic activity. Pace and Prairie (2005) and Abbasi and Chari (2008) offer four possible explanations germane to interpreting these results. First, the photochemical oxidation of inorganic or organic compounds consumes oxygen and cannot be discounted as a sink in these experiments. Nevertheless, cell growth was observed in the microcosm studies and the populations had shorter generation times in light versus dark incubations (11 and 14 days, respectively). Second, samples incubated in the light may experience higher incubation temperatures, but this possibility can be excluded because the dark samples were wrapped in aluminum foil and incubated under illumination, making uniform temperature differences between treatments unlikely. Finally, oxygen may be consumed by phototrophic algae and cyanobacteria through pseudocyclic electron transport (i.e., the Mehler reaction) or by the CO$_2$-fixing enzyme RuBisCO (ribulose-1,5-bisphosphate carboxylase/oxygenase) during photorespiration. These latter mechanisms provide the most plausible explanation for the elevated oxygen consumption rates during light incubation (Fig. 5B).



Meltwater production and duration in the ice are highly relevant to microbial processes in the WCA. Modeling of the ice properties under the local meteorological conditions from 2014 to 2016 suggests water content in the weathering crust increased gradually with the onset of the ablation season and that a substantial fraction of this water persisted in the ice for ~7.5 months each year (Fig. 1). Given that WCA cell populations had maximum

generation times of 2 weeks at the in situ temperature, the duration of the meltwater interval implies that 16 generations per year were possible. Assuming retention of the cells in the surface during summer (Irvine-Fynn et al., 2012), the potential for WCA communities to increase total biomass by four orders of magnitude (i.e., $2^{16}$) each melt season is a level of production that could significantly influence the carbon cycling budgets of ecosystems directly associated with the glacier and its drainage system (Anesio et al., 2010).

The availability of biologically labile organic carbon in WCA samples was supported by no observable lag in the time-dependent reduction of XTT to its formazan product (Fig. 5A), and also demonstrated the presence of cells with active respiratory chains and electron donors to drive oxidative phosphorylation. Elevated cell and ATP concentrations (Fig. 4A) were observed at depths with the highest predicted water contents (Fig. 1) and positively correlated with the concentration of bicarbonate (Fig. 3A), consistent with in situ microbial respiration

and growth in the WCA. Several horizons below predicted depths of meltwater production showed relative increases in the concentration of cells and ATP (i.e., 4.0 to 4.1, 6.3 to 6.4, and 8.3 m; Fig. 4A), but the microbial assemblages detected at these depths were distinctly different from those in the near-surface ice (Fig. 6A). Hence, we conclude that these represent cell populations that were deposited on the glacier's surface at a time in the past, became archived within the ice chronology, and glacier movement conveyed to the ablation zone. The positive

correlation of the cell and ATP data with $Al^{3+}$ and $K^+$ likely reflect contributions from aeolian transport to the ice surface and subsequent weathering, but the data for $Co^{2+}$ (Fig. 3A) are interesting when juxtaposed with the results of Taylor and Sullivan (2008). This study showed that uptake of $Co^{2+}$ and vitamin $B_{12}$ synthesis by bacteria in Antarctic sea ice supplied this vital micronutrient to algae auxotrophic for $B_{12}$. Consequently, the higher concentrations of $Co^{2+}$ observed in the weathering crust (Fig. 3A; Supplementary Table 2) could be associated

with community micronutrient recycling. Interactions between phototrophic and heterotrophic guilds in this sympagic environment should be enhanced by the close proximity offered on cell-particle aggregates (Fig. 4B), and those observed shared characteristics similar to those reported in the ice covers of Antarctic dry valley lakes (Priscu et al., 1998), cryoconite holes (Hodson et al., 2010; Langford et al., 2010), and sea ice (Riebesell et al., 1991).

Surface water transport and melting of glacial ice are the processes most likely providing microbial inoculum to the WCA. According to NMDS ordination, microbial assemblages in the deepest boreholes (i.e.,





originating from englacial ice) clustered by year and were highly significantly different from samples in the glacier's near-surface (Fig. 6A). Considering ablation rates of 4 to 15 cm day$^{-1}$ at the Matanuska Glacier during May to August (Ensminger et al., 1999; Reynolds, 2005; Mankoff and Russo, 2013), most of the ice column sampled in 2014 would have ablated by the following season (i.e., relative to the ice surface in 2014, the samples

collected in 2015 originated from a deeper horizon in the glacier). Curiously, data from a 10 m borehole (BH1) on two consecutive days in 2014 (BH1a and BH1b) clustered with near-surface ice samples, but the assemblages observed on the third day (BH1c) were more similar to those in the deeper 2014 boreholes (Fig. 6A). Movement of the glacier over irregular terrain has highly fractured the ice in the marginal area studied, and the hydrologically-connected fracture networks of temperate glaciers are the primary pathway for water transport to the bed (Fountain

et al., 2005). The temporal changes observed in BH1 support the possibility that microbes originating from deeper portions of the ice column can be mobilized and englacially transported with water on relatively short time frames.

Microbial communities in the 2014 and 2015 samples from shallow boreholes (4 m), a supraglacial stream, and the WCA formed a loose cluster in NMDS ordination space (Fig. 6A). Comparison of bulk meltwater (BH4a and BH10a) with water that subsequently percolated into the boreholes (BH4b, BH10b, BH10c, and

BH10d) indicated no significant difference in community structures (p = 0.138, n = 6). The community in the first WCA sample collected in 2015 (BH10b) was considerably less diverse than that in the borehole meltwater (BH10a; Fig. 4C), plotted on the periphery of the near-surface cluster in NMDS analysis (Fig. 6A), and had a structure significantly different from the other WCA samples (p = 0.015, n = 6). However, samples collected from BH10 over the following two days (BH10c and BH10d) contained more diverse communities (Fig. 4C) that were

more similar to each other and the 2014 WCA sample BH4b. Sample BH10b was collected after a period of rainfall, contained a cell concentration 5- to 14-fold higher than the other WCA samples (Fig. 4C), and was highly enriched with two gammaproteobacterial OTUs in the genera *Pseudomonas* and *Stenotrophomonas* that comprised 54% of the sequences. Together, the data from BH10 provide anecdotal evidence for a hydrologic response to precipitation that enhanced the mobilization of microbes from other hydrogeological regions and/or

facilitated the transfer of species deposited in rain to the WCA.

If the WCA represents an ecotone between the supraglacial and englacial environment, then the presence of distinct species and community structures would be expected. However, a strict interpretation of the WCA community composition based on data from the percolating water may be complicated by the co-occurrence of microbes in the samples that originated from aquatic ecosystems on the surface, glacial ice melt, and wet or dry

deposition. Odds ratios derived from sequence abundances (Fig. 6B) indicate that 23 OTUs affiliated with eight bacterial phyla and the plastids of golden and green algae (Supplementary Fig. 2) were enriched >five-fold in the





WCA relative to englacial ice samples. Four of these OTUs (OTU21, 31, 48, and 53) were at abundances >1% of the sequences in the WCA and near-surface ice samples but represented <0.1% of the sequences in supraglacial water, suggestive for the presence of distinct OTUs in the weathering crust. The largest odds ratios were observed for deltaproteobacterial OTUs in the 2014 samples, but three OTUs with high identity to plastid 16S rRNA gene

sequences of Zygnematophyceae and Chrysophyceae taxa (OTU8, 9, and 48) had odds ratios that ranged from 5 to 35 during both years (Fig. 6B) and represented 11 to 16% of sequences detected in the near-surface ice samples (BH3, BH4a, BH5, BH9, and BH10a). Phylogenetic analysis of 18S rRNA gene sequences amplified from a WCA sample revealed OTUs closely related to snow and ice algae in the genera *Ancylonema, Mesotaenium,* and *Ochromonas* (Fig. 6C; Supplementary Table 4). Previous studies have identified *Ancylonema nordenskioldii* and

*Mesotaenium breggrenii* in Matanuska Glacier cryoconites near the terminus (Takeuchi et al., 2003) as well as on the surface of the Greenland and Antarctic ice sheets (Anesio et al., 2017; Lutz et al., 2018).

## 5 Conclusions

During the ablation season, microbes in the glacial weathering crust are provided with liquid water, solar radiation, oxygen, and conditions suitable for biogeochemical processing. Though phototrophic activity contributes new

organic carbon and generates oxygen to drive heterotrophic metabolisms, primary producers and consumers in the WCA must acquire other vital nutrients to support growth, which likely occurs via co-transport with percolating surface water and release from melting glacial ice. Decreased solar irradiance and temperature at the onset of winter constrains ecosystem processes by freezing water in the WCA, entrapping the microbial community in a near-surface refuge. Surviving fractions of these populations then serve as inocula for the

communities that develop in the subsequent melt season. Ice-water interfaces in the weathering crust promote downward light scattering (Gardner and Sharp, 2010) and microbial growth on glacier surfaces leads to "biological darkening" of ice (Cook et al., 2017; Stibal et al., 2017; Tedstone et al., 2017). As both of these processes increase melt rates and may lead to a positive feedback that accelerates further meltwater production, the physical and biological contributions of glacial weathering crusts to albedo reduction warrants more detailed investigation.

*Data availability*: The 16S rRNA gene sequence data are deposited in the National Center for Biotechnology Information's Sequence Read Archive (project number SRP130982). The cloned 18S rRNA gene sequence data are assigned the GenBank accession numbers MH037315 to MH037321.

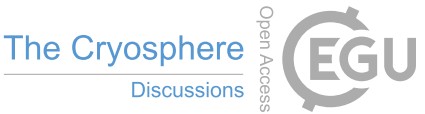

*Supplement link*: The Supplement related to this article is available online at: TBD

*Author contributions*: B.C.C., S.M.T., P.T.D., and W.C.S. designed the research; B.C.C., H.F.L., C.L.D., E.E.O., S.U.N., K.F.M., S.M.T., and P.T.D. performed the research; B.H. conducted analytical analysis; W.C.S.
developed and provided technological resources; B.C.C., C.L.D., S.M.T., and P.T.D. analyzed the data; and B.C.C. wrote the manuscript with contributions from all authors.

*Competing interests*: The authors declare no conflict of interest.

*Acknowledgements*: This research was supported by a grant from the NASA ASTEP Program (NNX11AJ89G). Partial support was also provided by the Institute of Food and Agricultural Sciences at the University of Florida. We thank J. Farrar for assistance with extracting samples for nucleic acid analysis and J. Cook, A. Hodson, and J. Priscu for discussion. Support and assistance provided by the following 2014 and 2015 VALKYRIE field team members was crucial to the success of this research: N. Bramall, E. Clark, C. Flesher, J. Harman, B. Hogan, J.
Moor, D. Rickel, D. Sampson, V. Siegel, and V. Yuan.

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







**Figure 1.** Temperature and ice water content during 2014 to 2016 (A) Modeled ice temperature was calculated using a finite



difference solver to a one-dimensional diffusion-advection-production equation. Only values between 0 (red limit) and -5°C (blue limit) are plotted. (B) The model is driven by the ERA-interim reanalysis 2 m-air temperature (black dots) and downward radiation data (brown dots) for the region (Dee et al., 2011). Both forcings are plotted at hourly intervals after smoothing with a daily (surface temperature) and weekly (radiation) running-average filter to suppress short-term variability. The greyed

5   region in panel B represents the summer 2015 interval plotted in panel C when ice temperature data were collected. The arrows in panel A and B indicate the timing of the two field campaigns. (C) Comparison between ice temperatures observed with a temperature string deployed into a 45 m-deep borehole in July and August of 2015 (solid lines) and the modeled ice temperatures (dashed lines).





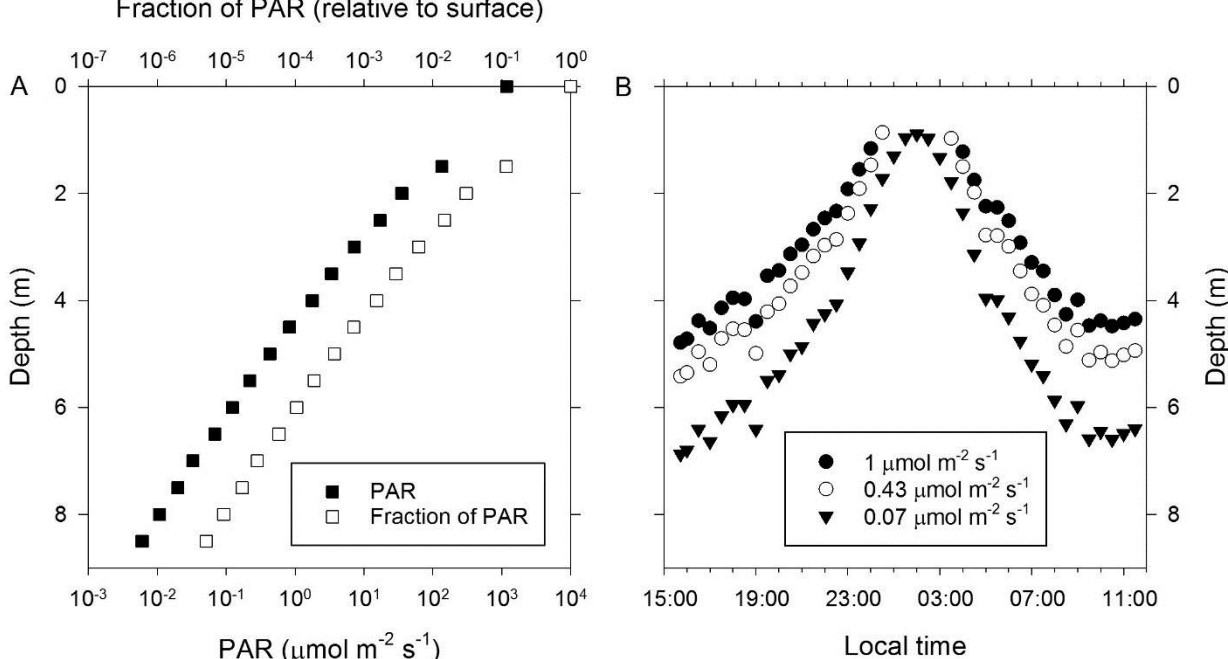

**Figure 2.** PAR optical properties of the near-surface ice during the 2014 ablation season. (A) Measurement of PAR attenuation to a depth of 8.5 m. (B) Diel pattern for select PAR fluxes that support oxygenic photosynthesis in cold and shaded aquatic environments (Priscu et al., 1988; Hawes and Schwarz, 2001). The data plotted were derived from an overnight measurement
5    with the sensor at a depth of 3.5 m and then using data from (A) to extend the overnight profile to the upper 8.5 m ice column.



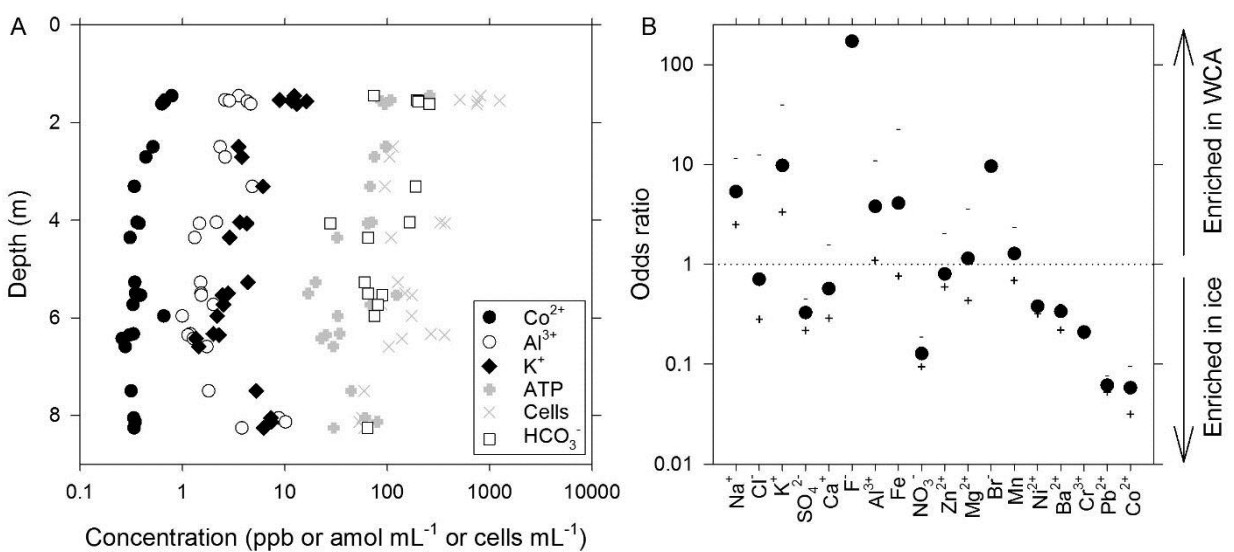

**Figure 3.** Geochemical composition of melted ice and water from the WCA. A) Depth profile for analytes correlated with the cell and/or ATP data. B) Ratio of compounds in the WCA water samples relative to the average in ice from depths of 1.4 to 11.3 m. The order of compounds is based on their abundance in the WCA sample (i.e., decreasing abundance from left to right). The plus and minus symbols are the ratios based on the maximum and minimum values, respectively, observed in the ice profile.



**Figure 4.** Cells and biomass in samples of melted ice and water from the WCA. A) Concentration of cells and ATP from depths of 1.4 to 11.3 m. B) Epifluorescent micrograph showing example of cell-particle aggregates observed in samples from the glacier's near-surface (sample BH4a). C) Comparison of community diversity (open circles) and cell concentration data from the ice (white bar) with water that percolated into the borehole (grey bars). The ice value was derived by averaging the data plotted in panel A. The inverse Simpson index for the ice is based on average values from sampled depths between 2 to 15 m (BH6a, BH6b, BH6c, BH7a, BH7b, and BH7c). The error bars in panels A and C indicate the standard deviation from the mean.





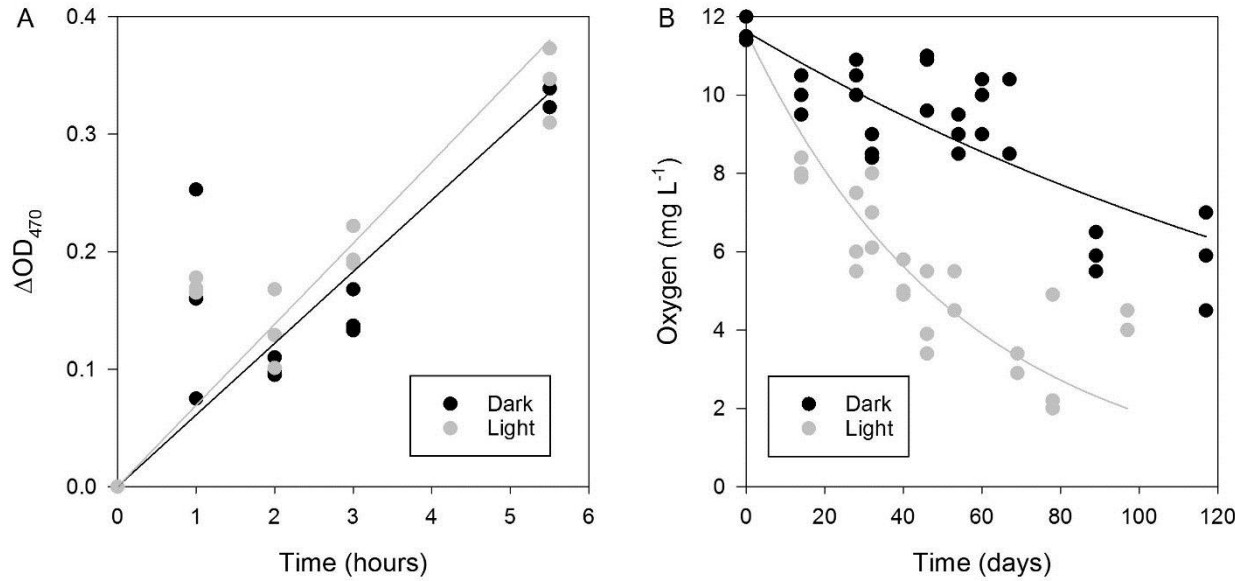

**Figure 5.** Respiratory activity of microorganisms in the WCA. (A) Time-dependent reduction of XTT for samples (BH5) incubated in situ. Linear regression models for the data are also plotted (B) Oxygen consumption in the light (~270 μmol photons $m^2$ $s^{-1}$) and dark over approximately 3 months at 5°C (sample BH10c). The data were fit with an exponential model

5 (Dark, y = 11.6$e^{-0.00553}$, $r^2$ = 0.676; Light, y = 11.6$e^{-0.0167}$, $r^2$ = 0.644).



**Figure 6.** Microbial community structure in the ice and WCA based on analysis of 16S/18S rRNA genes. (A) NMDS



ordination plot of communities observed in samples of englacial (upper left and right) and near-surface (bottom) ice. (B) Odds ratios for taxa enriched in the near surface ice samples. The OTUs shown have abundances >0.25% of all sequences and odds ratios >5-fold in samples from 2014 or 2015. (C) Phylogenetic analysis of amplified 18S rRNA gene sequences (positions 566 to 1200, *Saccharomyces cerevisiae* numbering) related to algae in Zygnematophyceae and Chrysophyceae classes. Genbank

5  accession numbers are listed in parentheses. The scale bar represents 0.020 substitutions per site and bootstrap values are shown as a percentage of 1000 replications.