# Peer review of "Microbial processes in the weathering crust aquifer of a temperate glacier"

_The Cryosphere, 2018_

## Referee Comment (RC1) · Anonymous Referee #1 · 4 Sep 2018

There is an increasingly amount of literature showing that microbial presence (particularly pigmented algae) on the surface of the ice can promote positive feedbacks between melting and microbial activity. There is also plenty of information on microbial processes in cryoconite holes. However, the microbial structure within the weathering crust aquifer (WCA) of glaciers has only been conceptually explored, and the reality is that there are hardly any data looking at microbial processes on glacial surfaces at an appropriate scale to improve our understanding of these processes in the WCA. I think that this paper goes a long way to explore microbial processes at the WCA, and it is completely appropriate for a journal like "Cryosphere". I really enjoyed reading it. The combined modelling, geochemical and microbial measurements makes the paper particularly attractive and informative.

[Figure]

My main suggestion for improvement is related to the presentation of the sampling procedure in the boreholes, which I think is a crucial part of the study. I am sure most readers would like to have a better visualisation of how samples were obtained by having some photos and/or a schematic figure (either in the main paper or as supplementary information). I have no doubt that the study has captured well the weathering crust microbial processes. The data of microbial composition provides some solid evidence of a different community in the WCA compared to the surface community, and some more information in the methods would help to make this point clear.

Minor suggestions: Page 4, line 16 – Please specify how the boreholes were monitored. I think this will also help to clarify the first paragraph of the results section.

Page 8, line 18 – please specify the volume of the serum vials.

Last paragraph of page 17 – Maybe it is worthy to mention/reinforce the fact that the sample used for those incubations has a relatively high proportion of Cyanobacteria/plastids. Nevertheless, the cell number increase during the oxygen consumption experiment indicates a strong bottle effect during the incubations. It would be good if this can be further discussed in line with the calculations in the first paragraph of page 18.

The literature on microbial processes at the very ice surface provides quite often evidence for organic carbon accumulation, which in turn results in the darkening of the ice. Is it possible for the authors to make inferences (based on the incubation experiments and microbial community composition of the WCA) whether the WCA microbial processes could have a role on surface activity (e.g., via recycling of nutrients that become available to surface organisms as the ice ablates exposing WCA communities at the surface)?

---

## Referee Comment (RC2) · Anonymous Referee #2 · 9 Sep 2018

General comments: The authors present a detailed study of near-surface ice of a temperate glacier in Alaska. The specific focus is the microbial habitat or ecotone that this near-surface environment may represent. Numerous recent studies have now begun exploring this, conceptually or otherwise, but this is the first paper to conclusively demonstrate this environment is active for a temperate glacier surface. Data include observations and modelled estimations of ice temperature, biogeochemical measurements and microbial community composition. The paper concludes that the weathering crust that the near-surface represents is a viable location for microbial activity, and estimates of biomass growth are made. The paper is well written in the main, and methods are fully detailed. There are no critical experimental flaws to note, and data presented appear robust and sound. Figures are fair and reasonable in general. Some aspects

of the introduction and discussion/conclusion could perhaps be strengthened to help benefit the paper's narrative, focus and impact. That said, the paper was a constructive and informative read.

Specific comments: The more significant observation is that the use of borehole thermistor data from 15m to 45m seems to be somewhat inappropriate. This is outside the zone of interest, and the discussion of these results alludes to weaknesses in both the thermistor data itself and the modelled ice temperature, at least to make confidence in both aspects rather clouded. This detracts and confuses the paper, and takes the reader into areas that simply add little to the focus of the paper. Recommendation would be to simply use the modelled surface temperature profiles to 15m as a proxy for the temperature conditions. The other thermistors lie outside this, so while useful to test the broader scale applicability of the thermal model, the deeper measurement points can not really be used to validate the near-surface model, particularly with the uncertainties discussed, and so seem to be simply adding data unnecessarily (nor essential) to the paper. Suggest removal of all information on the uncertain borehole data. Simplify this to the use of the model as a proxy estimate for the surface conditions. Discussion can then allude to the need to better characterise the thermal behaviour of the WCA, and instrument the uppermost few metres, including adding the potential influence of snow cover on thermal regime over the full annual cycle. P2: Opening paragraph, seems to slightly confuse the goal and focus of the study, and could be seen as rather weak and less targeted than perhaps might be achieved. Recommend revisiting and reworking. Now, the concept of glacier surfaces as an ecosystem (e.g. Hodson et al., 2008; Stibal et al., 2012, Nature Geoscience; Hotaling et al., 2017, Environmental Microbiology) is well-accepted, and the references particularly to firn storage rather distract from the core topic here which is the shallow near-surface ice in the ablation zone. Preference might be to keep the study's focus clear from the outset to better guide the reader. Perhaps better to consider Cooper et al. (2018, Cryosphere) and Smith et al. (2017, PNAS) who focus on water storage in bare ice in Greenland, or delayed runoff in mountain glacier settings (Munro, 2011, Hydrological Processes).

Consider opening with need to understand the ice surface as a locus of biological activity, relevant with recognition of water storage and delay. This would better guide the reader into the material that follows. L25: Perhaps reference to earlier LaChapelle (1950s) work on this, and Munro's (1990) examination of subsurface melting would be appropriate to evidence knowledge of this phenomenon. Similarly, Muller and Keeler (1969) were the first to use "weathering crust" specifically for glacier ice, although others used ablation rind, honeycomb ice, ablation crust etc. (see earlier discussion texts regarding glacier ice structure) – perhaps revise citation position to highlight and better evidence this and source of the terminology. P3 L2: Particularly for temperate glaciers – could be emphasised, given this is one aspect of novelty here. Note, here, given Larson's (1977, 1978) studies on water balance and meltwater storage in the near-surface of an Alaskan glacier, it is surprising these references are absent, and they could be helpful for defining or better justifying the depth range of of WCA zone examined and sampled here. See earlier comment regarding maintaining clear focus in the opening sections. See also Fountain and Walder's (1998) comments on this ice 'zone'. P4 L14: The authors hoefully should be aware of Stevens et al. (2018, Hydrological Processes) as well as Cooper et al. (2018, Cryosphere) – perhaps here, and/or elsewhere, recognition could be given to work examining porosity and permeability of the WCA. This is a current topic, and updates to sources cited could be included to keep the paper contemporary. P5 L23: Perhaps missed something in this section, but the "water content" aspect isn't perhaps as clear as could be effected. This section seems to describe temperature in detail, but does not quite give a sufficiently direct indication of water content calculations, especially if calculated for the model domain rather than the porous layer itself. Suggest clarification for readers less familiar with thermal models for ease of accessibility, given water volume is later reported in figures. L5 L24: For the WCA, typically only a few m, the use of a 45 m borehole seems to be excessive and rather misaligned with the zone of interest (see earlier comment). Similarly the model of 100 m seems to lie outside the region of observations. Note misalignment of information between the 45 m borehole and the 20 m of thermal data presented, update

caption to reflect this – and perhaps reflecting earlier suggestion of reducing the use of less certain thermal data. While use of deeper temperature records to seek to validate the thermal model is admirable, there seem to be significant uncertainties both in the model and the thermistor data that render this approach equivocal and conjectural. See comment above, which may help here. P16 L20: Fountain and Walder (1998, Reviews Geophysics) discuss the WCA and Irvine-Fynn et al. (2011) phrase this as a "perched aquifer". Also see Stevens et al. (2018, Hydrological Processes), Cooper et al. (2018, Cryosphere), and Smith et al. (2017, PNAS). L27: Consider revising as the WCA may be shallow (see LaChapelle's earlier work), and while lower density (by volume) ice may exist at the surface, typically the water table lies below this. See Muller and Keeler's (1969) conceptual diagram of this, and as evidenced by water depths in cryoconite holes. It would therefore be unlikely that ice would show a 50% liquid water by volume, rather this might be the very weathered surface ice, from which melt drains to the denser near-surface. Similarly, the Antarctic comparison seems to understate the thermal conditions both at surface and subsurface, which would influence WCA development. Recommend adjusting the wording here to reflect this and making comparisons more relatable. P18 L4: given the temperature model excludes snow cover (see Methods description), is the 7.5 months realistic, and does this tally with in situ snow cover, which itself will affect WCA development and closure? Perhaps revise values here to reflect this, or caveat that this neglects the influence of snow on the WCA – a process yet to be reported. P20 L12: This section becomes more conjectural and loses focus on content of paper. Suggest revising conclusion to reflect the data and relationships shown. Nothing here directly reports on biological influence on the albedo of glaciers. Rather, identification of a functional microbial environment on a temperate glacier – which supports ideas and aspects revealed in Arctic and ice-sheet settings – is important. Note, there are perhaps two 'definitions' of "biological darkening": one potentially introduced in Irvine-Fynn et al. (2012, Environmental Microbiology) referring to microbe-mineral retention; and secondly, the more recent phrasing used when referring to the 'bio-albedo' and apparent darkening of some ice surfaces by active

biomass (e.g. van den Broeke et al, 2017, Current Climate Change Reports) which was termed "biotic acceleration of glacier melt" (Koshima et al., 1993, IAHS). However, the processes underlying these definitions are subtly different. Importantly, none of the references cited specifically use the phrase "biological darkening". Much of the bioalbedo is related to ice surface algae, not necessarily the same community as that in the WCA. Please use quotations correctly, and recommend retaining focus on actual data and findings of the paper rather than seeking to link to other topic areas.

Technical comments: A number of more stylistic, but nonetheless important observations include: P2 L20: Scott et al. (2010, Ann Glaciology) show microbial nutrient turnover in supraglacial streams, which may be relevant here. L21: Please use multiplication not the letter 'x'. Noted elsewhere throughout (e.g. P6 L14). P6 L8: Check journal style, but perhaps revise unit to $^\circ$C. P7 L3: Confirm KPAR is K in equation. P9 L9: Check journal style for unit / constant here. P10 L30: Define rs and rp here, as used elsewhere in Results section. P11: Results, please check stylistics as throughout there are contrasting uses (or absences) of '0' before decimal points. A pet peeve is some journals/publishers that have removed zeros from quantities – whether numeric measurements or statistical values. P11 L17: revise and condense. Near-surface is unlikely, by definition, to be 45 m at depth. P12 L15: Slight repetition from Methods. Suggest simply presenting equation in methods including citation, and referring to this here with result. P12 L19: See Larson (1970s) references to support this depth of WCA or photic zone, if thermally still below zero. Consider further in discussion. P13 L12: see earlier comment re. definition of correlation coefficients. Condense. P14: Check style for p-values. Italic elsewhere. See also P19. P17 L11: unit consistency, elsewhere L/m2, later here, use of superscript negatives for 'per'. Recommend check and edit. P17 L13: Repetition of P12 L11-20, and immediately above. Revise, avoid repetition. P18: L7: Again, perhaps see Stevens et al. (2018). L18: useful to consider Xiang et al. (2009, FEMS Microbiology Ecology) as this is not altogether a new concept and has been discussed in the literature. L30: "emergence of ice in the actively melting ablation area" might be a stronger phrasing. P19 L9: Unclear why access to

bed is relevant here, suggest simply noting the fracture networks present in temperate ice may provide an explanation for near-surface to englacial linkage might be sufficient, particularly for emergent ice in the ablation area. L31: use "more than" rather than > here.

---

## Author Comment (AC1) · 29 Oct 2018

*Please note pages/lines referenced by the authors are to revised text*

1. Comments from Referee: "My main suggestion for improvement is related to the presentation of the sampling procedure in the boreholes, which I think is a crucial part of the study. I am sure most readers would like to have a better visualisation of how samples were obtained by having some photos and/or a schematic figure (either in the main paper or as supplementary information). I have no doubt that the study has captured well the weathering crust microbial processes. The data of microbial composition provides some solid evidence of a different community in the WCA compared to the surface community, and some more information in the methods would help to make

this point clear".

Author's response: We agree with this suggestion.

Author's changes in manuscript: A new figure has been included in the revised manuscript (Supplementary Figure 1 in revision) that includes a schematic of the approach we used for sampling and an image of a water sample being retrieved from one of the boreholes.

2. Comments from Referee: "Page 4, line 16 – Please specify how the boreholes were monitored. I think this will also help to clarify the first paragraph of the results section."

Author's response: Science personnel were on site daily from approximately 9 AM to 6 PM local time (i.e., during the diurnal peak of surface melt water production). The content of each drained borehole was examined daily (detailed inspection each morning after arriving on site and then periodically through the day). During the 2014 and 2015 sampling periods when it was verified that water was percolating laterally and collecting in the boreholes, each was monitored at no less than hourly intervals when personnel were on site.

Author's changes in manuscript: Additional information about how the boreholes were monitored has been added to the text (Pg 4, lines 18-22).

3. Comments from Referee: "Page 8, line 18 – please specify the volume of the serum vials."

Author's response: The water was collected in 30 mL serum vials

Author's changes in manuscript: This information has been added to the text (Pg 8, line 23).

4. Comments from Referee: "Last paragraph of page 17 – Maybe it is worthy to mention/reinforce the fact that the sample used for those incubations has a relatively high proportion of Cyanobacteria/plastids. Nevertheless, the cell number increase during

the oxygen consumption experiment indicates a strong bottle effect during the incubations. It would be good if this can be further discussed in line with the calculations in the first paragraph of page 18."

Author's response: These important points were not sufficiently emphasized in the manuscript.

Author's changes in manuscript: In the revision, the high abundance of phototrophic taxa inferred in the near-surface samples is referenced directly with respect to the oxygen consumption results (Pg 17, lines 29-30). We also revised the text to acknowledge that ex situ assays inevitably involve sampling disturbances and may provide conditions that allow microorganisms to reproduce at higher than in situ rates (i.e., the "bottle effect"; Pg 18, lines 9-10).

5. Comments from Referee: "The literature on microbial processes at the very ice surface provides quite often evidence for organic carbon accumulation, which in turn results in the darkening of the ice. Is it possible for the authors to make inferences (based on the incubation experiments and microbial community composition of the WCA) whether the WCA microbial processes could have a role on surface activity (e.g., via recycling of nutrients that become available to surface organisms as the ice ablates exposing WCA communities at the surface)?"

Author's response: While we mention how biogeochemical processes on the ice surface may influence those occurring in the WCA, the reverse scenario was not mentioned.

Author's changes in manuscript: A sentence has been added to the concluding paragraph (Pg 20, lines 26-27) stating the possibility that biomass turnover in the WCA may also mineralize and mobilize nutrients that fertilize biological activities on the ice surface.

---

## Author Comment (AC2) · 29 Oct 2018

*Please note pages/lines referenced by the authors are to revised text*

1. Comments from Referee: "The more significant observation is that the use of borehole thermistor data from 15m to 45m seems to be somewhat inappropriate. This is outside the zone of interest, and the discussion of these results alludes to weaknesses in both the thermistor data itself and the modelled ice temperature, at least to make confidence in both aspects rather clouded. This detracts and confuses the paper, and takes the reader into areas that simply add little to the focus of the paper. Recommendation would be to simply use the modelled surface temperature profiles to 15m as a proxy for the temperature conditions. The other thermistors lie outside this, so while

useful to test the broader scale applicability of the thermal model, the deeper measurement points can not really be used to validate the near-surface model, particularly with the uncertainties discussed, and so seem to be simply adding data unnecessarily (nor essential) to the paper. Suggest removal of all information on the uncertain borehole data. Simplify this to the use of the model as a proxy estimate for the surface conditions. Discussion can then allude to the need to better characterise the thermal behaviour of the WCA, and instrument the uppermost few metres, including adding the potential influence of snow cover on thermal regime over the full annual cycle."

Author's response: We thank the reviewer for thinking carefully about the relationship between modeled and observed temperatures. We have removed the 45 m temperature record as it does seem to detract from the main points we make in the paper. Moreover, it appears that we need to do a better job at explaining in the manuscript the take-away messages from both of these approaches. The output from our heat diffusion-production model is used in this manuscript to illustrate how subsurface temperatures and water content would evolve if the near-surface part of a glacier would behave as a homogeneous ice mass subjected to realistic surface temperature and radiation forcings. The model output is clearly shown in our figure (Fig. 1A) before it is used to compare model predictions with field temperature observations (Fig. 1C). Hence, readers interested in predictions of our idealized model can easily understand them before moving onto the part where we compare modeled and observed ice temperatures.

The field observations of temperatures at different depth demonstrate that, in reality, our study site is heterogeneous and allows processes other than heat diffusion and penetration of sunlight to impact ice temperatures. Our field temperature data suggest that the processes perturbing temperature distribution in a heterogeneous way involve vertical transport and refreezing of water (to explain observations of ice temperatures warmer than predicted by the model at a given depth) as well as flow of cold air through fractures and/or conduits (to explain observations of ice temperatures colder than predicted by the model at a given depth).

The mismatch between the observed and modeled temperatures does not represent noise that detracts from the focus of this paper but does represent by itself an insightful scientific finding. Basically, our model output represents a hypothesis and our field observations allow us to test this hypothesis. In this case we reject the hypothesis of homogeneous subsurface and speculate which features of the real system can explain the mismatch between model and observations. Future work by our team, or other researchers, may enable modeling of the effects of vertical water and air transport through fractures and conduits. However, at the present time we simply do not have sufficient data to treat these effects in a realistic, quantitative manner. Similarly, we do not have the data that would justify inclusion of a snow cover into our model. Generally, such a snow cover should produce two competing effects: (1) it will thermally insulate the underlying ice hindering its winter cooling, but (2) it will also hinder internal solar heating of the ice itself until the winter snow cover is completely melted. The two effects may largely cancel each other and should not have first-order impacts on the key results of our model.

Author's changes in manuscript: We removed the 45m temperature record from Figure 1C and changed the discussion of these results (Pg 12, lines 1-15).

2. Comments from Referee: "P2 Opening paragraph, seems to slightly confuse the goal and focus of the study, and could be seen as rather weak and less targeted than perhaps might be achieved. Recommend revisiting and reworking. Now, the concept of glacier surfaces as an ecosystem (e.g. Hodson et al., 2008; Stibal et al., 2012, Nature Geoscience; Hotaling et al., 2017, Environmental Microbiology) is well-accepted, and the references particularly to firn storage rather distract from the core topic here which is the shallow near-surface ice in the ablation zone. Preference might be to keep the study's focus clear from the outset to better guide the reader. Perhaps better to consider Cooper et al. (2018, Cryosphere) and Smith et al. (2017, PNAS) who focus on water storage in bare ice in Greenland, or delayed runoff in mountain glacier settings

(Munro, 2011, Hydrological Processes). Consider opening with need to understand the ice surface as a locus of biological activity, relevant with recognition of water storage and delay. This would better guide the reader into the material that follows."

Author's response: These are excellent suggestions.

Author's changes in manuscript: The discussion of firn aquifers has been eliminated and replaced with text describing how the weathering crust influences the timing and magnitude of meltwater delivery to supraglacial, subglacial, and proglacial systems (Pg 2, lines 7-10).

3. Comments from Referee: "L25: Perhaps reference to earlier LaChapelle (1950s) work on this, and Munro's (1990) examination of subsurface melting would be appropriate to evidence knowledge of this phenomenon. Similarly, Muller and Keeler (1969) were the first to use "weathering crust" specifically for glacier ice, although others used ablation rind, honeycomb ice, ablation crust etc. (see earlier discussion texts regarding glacier ice structure) – perhaps revise citation position to highlight and better evidence this and source of the terminology."

Author's response: We thank the reviewer for providing information on the earliest observations that shaped understanding on the structure, formation, and properties of the weathering crust.

Author's changes in manuscript: The LaChapelle (1959) and Munro (1990) papers are cited in the revision and the text has been edited to unambiguously attribute the term "weathering crust" to Müller and Keeler (1969) (Pg 2, lines 26-28).

4. Comments from Referee: "P3 L2: Particularly for temperate glaciers – could be emphasised, given this is one aspect of novelty here. Note, here, given Larson's (1977, 1978) studies on water balance and meltwater storage in the nearsurface of an Alaskan glacier, it is surprising these references are absent, and they could be helpful for defining or better justifying the depth range of WCA zone examined and sampled here. See

earlier comment regarding maintaining clear focus in the opening sections. See also Fountain and Walder's (1998) comments on this ice 'zone'."

Author's response: That there have been no studies of weathering crusts in temperate glaciers is a point worth emphasizing. The intention of the paragraphs' final sentence is to stress the dearth of information on biogeochemical processes in the near-surface aquifers of any glacier. As such, we believe that a description of water balance and storage for other Alaskan glaciers moves this discussion off point.

Author's changes in manuscript: The final paragraph in this sentence has been edited to mention that no data has been available on the weathering crust ecosystems of temperature glaciers (Pg 3, lines 2-5).

5. Comments from Referee: "P4 L14: The authors hoefully should be aware of Stevens et al. (2018, Hydrological Processes) as well as Cooper et al. (2018, Cryosphere) – perhaps here, and/or elsewhere, recognition could be given to work examining porosity and permeability of the WCA. This is a current topic, and updates to sources cited could be included to keep the paper contemporary."

Author's response: Agreed but such a discussion is not appropriate in the methods section.

Author's changes in manuscript: The porosity and permeability observations of Cooper et al. and Stevens et al. on the WCA are now referenced in the introduction (Pg 2, lines 7-9) and discussed on Pg 16, lines 26-29.

6. Comments from Referee: "P5 L23: Perhaps missed something in this section, but the "water content" aspect isn't perhaps as clear as could be effected. This section seems to describe temperature in detail, but does not quite give a sufficiently direct indication of water content calculations, especially if calculated for the model domain rather than the porous layer itself. Suggest clarification for readers less familiar with thermal models for ease of accessibility, given water volume is later reported in figures."

Author's response: We thank the referee for catching the fact that our methods section did not include sufficiently detailed explanation of our approach towards calculating internal melt water fraction.

Author's changes in manuscript: We added a few sentences that explain how internal water content is calculated in the model (Pg 6, lines 16-19).

7. Comments from Referee: "L5 L24: For the WCA, typically only a few m, the use of a 45 m borehole seems to be excessive and rather misaligned with the zone of interest (see earlier comment). Similarly the model of 100 m seems to lie outside the region of observations. Note misalignment of information between the 45 m borehole and the 20 m of thermal data presented, update caption to reflect this – and perhaps reflecting earlier suggestion of reducing the use of less certain thermal data. While use of deeper temperature records to seek to validate the thermal model is admirable, there seem to be significant uncertainties both in the model and the thermistor data that render this approach equivocal and conjectural. See comment above, which may help here."

Author's response: As mentioned above, we think it is important to keep some of the comparison of observed temperatures to modeled temperatures. We have followed referee's suggestion and deleted the 45 m record, which was the most different from its simulated equivalent. However, we do think that it is important to retain the comparison between the temperatures at the other depths as they underline the point that the upper part of the glacier is highly heterogeneous in reality and enables vertical advection of heat, water, and microbial matter. We see no benefit in limiting the discussion to just the results of simulations.

Author's changes in manuscript: We have deleted the 45m temperature record from Figure 1C and changed the results section accordingly (Pg 12, lines 1-15).

8. Comments from Referee: "P16 L20: Fountain and Walder (1998, Reviews Geophysics) discuss the WCA and Irvine-Fynn et al. (2011) phrase this as a "perched aquifer". Also see Stevens et al. (2018, Hydrological Processes), Cooper et al. (2018,

Cryosphere), and Smith et al. (2017, PNAS)."

Author's response: We are aware of prior classification of the WCA as a perched aquifer but are confused about the intention of this comment.

Author's changes in manuscript: None.

9. Comments from Referee: "L27: Consider revising as the WCA may be shallow (see LaChapelle's earlier work), and while lower density (by volume) ice may exist at the surface, typically the water table lies below this. See Muller and Keeler's (1969) conceptual diagram of this, and as evidenced by water depths in cryoconite holes. It would therefore be unlikely that ice would show a 50% liquid water by volume, rather this might be the very weathered surface ice, from which melt drains to the denser near-surface. Similarly, the Antarctic comparison seems to understate the thermal conditions both at surface and subsurface, which would influence WCA development. Recommend adjusting the wording here to reflect this and making comparisons more relatable."

Author's response: We agree with these suggestions.

Author's changes in manuscript: The sentence discussing the unlikely possibility of water contents as high as 50% has been eliminated and we have edited the comparison to Antarctic weathering crusts by pointing out that the ice is much colder and that the penetrating radiation is mostly used to warm the ice rather than melt it (Pg 16 lines 25-29).

10. Comments from Referee: "P18 L4: given the temperature model excludes snow cover (see Methods description), is the 7.5 months realistic, and does this tally with in situ snow cover, which itself will affect WCA development and closure? Perhaps revise values here to reflect this, or caveat that this neglects the influence of snow on the WCA – a process yet to be reported."

Author's response: This is an important point that we now discuss. We do not have the

data to explicitly treat the problem so we discuss how inclusion of snow cover could change our results.

Author's changes in manuscript: We have included a discussion of this on Pg 18 (lines 4-9).

11. Comments from Referee: "P20 L12: This section becomes more conjectural and loses focus on content of paper. Suggest revising conclusion to reflect the data and relationships shown. Nothing here directly reports on biological influence on the albedo of glaciers. Rather, identification of a functional microbial environment on a temperate glacier – which supports ideas and aspects revealed in Arctic and ice-sheet settings – is important. Note, there are perhaps two 'definitions' of "biological darkening": one potentially introduced in Irvine-Fynn et al. (2012, Environmental Microbiology) referring to microbe-mineral retention; and secondly, the more recent phrasing used when referring to the 'bio-albedo' and apparent darkening of some ice surfaces by active biomass (e.g. van den Broeke et al, 2017, Current Climate Change Reports) which was termed "biotic acceleration of glacier melt" (Koshima et al., 1993, IAHS). However, the processes underlying these definitions are subtly different. Importantly, none of the references cited specifically use the phrase "biological darkening". Much of the bioalbedo is related to ice surface algae, not necessarily the same community as that in the WCA. Please use quotations correctly, and recommend retaining focus on actual data and findings of the paper rather than seeking to link to other topic areas."

Author's response: We agree with this suggestion.

Author's changes in manuscript: The text referring to biological darkening has been removed and the conclusion section has been edited to focus the data presented (Pg 20-21, lines 25-29, 1-4).

12. Comments from Referee: "P2 L20: Scott et al. (2010, Ann Glaciology) show microbial nutrient turnover in supraglacial streams, which may be relevant here."

Author's response: Agreed.

Author's changes in manuscript: The observations of Scott et al. are now discussed in this section (Pg 2 lines 17-20).

13. Comments from Referee: "L21: Please use multiplication not the letter 'x'. Noted elsewhere throughout (e.g. P6 L14)."

Author's response: Agreed.

Author's changes in manuscript: The multiplication symbol now replaces "x" here and throughout the revised text.

14. Comments from Referee: "P6 L8: Check journal style, but perhaps revise unit to âŮęC."

Author's response: Agreed.

Author's changes in manuscript: The text has been revised to present the value as "mm oC-1 d-1".

15. Comments from Referee: "P7 L3: Confirm KPAR is K in equation."

Author's response: Yes, KPAR is K in the equation and the confusion is understandable.

Author's changes in manuscript: The text has been revised to make KPAR/K unambiguous (Pg 7 lines 11-12).

16. Comments from Referee: "P9 L9: Check journal style for unit / constant here."

Author's response: None

Author's changes in manuscript: The "x" has been replaced with the multiplication symbol.

17. Comments from Referee: "P10 L30: Define rs and rp here, as used elsewhere in Results section."

Author's response: None

Author's changes in manuscript: Designations for the Pearson's and Spearman's correlation coefficients now appear on Pg 11 lines 3-4.

18. Comments from Referee: "P11: Results, please check stylistics as throughout there are contrasting uses (or absences) of '0' before decimal points. A pet peeve is some journals/publishers that have removed zeros from quantities – whether numeric measurements or statistical values."

Author's response: By convention, the zero before the decimal point of correlation coefficients are typically omitted. However, we are not married to this convention.

Author's changes in manuscript: A zero had been added before the decimal for correlation coefficients reported throughout the manuscript.

19. Comments from Referee: "P11 L17: revise and condense. Near-surface is unlikely, by definition, to be 45 m at depth."

Author's response: See our response to comment #1. We have removed the 45 m temperature record.

Author's changes in manuscript: This section has been renamed as "Physical conditions" (Pg 11, line 20).

20. Comments from Referee: "P12 L15: Slight repetition from Methods. Suggest simply presenting equation in methods including citation, and referring to this here with result."

Author's response: Agreed

Author's changes in manuscript: The text has been edited to eliminate redundancy with the methods (Pg 12 line 19).

21. Comments from Referee: "P12 L19: See Larson (1970s) references to support

this depth of WCA or photic zone, if thermally still below zero. Consider further in discussion."

Author's response: This sounds like a relevant paper to read and potentially cite; however, we are not sure of the study that the reviewer is referring to.

Author's changes in manuscript: None

22. Comments from Referee: "P13 L12: see earlier comment re. definition of correlation coefficients. Condense."

Author's response: We have made this change (see response above).

Author's changes in manuscript: A zero had been added before the decimal for correlation coefficients reported throughout the manuscript.

23. Comments from Referee: "P14: Check style for p-values. Italic elsewhere. See also P19."

Author's response: Some p-values were not italicized in the initial version and we could not find guidelines for presenting p-values in TC.

Author's changes in manuscript: All p-values reported in the manuscript have been italicized for consistency.

24. Comments from Referee: "P17 L11: unit consistency, elsewhere L/m2, later here, use of superscript negatives for 'per'. Recommend check and edit."

Author's response: The conversion to superscript units makes this sentence quite awkward, i.e., "200 L of liquid water m-2 of glacier surface…". Hence, we have kept the original text.

Author's changes in manuscript: Figure 1 has been edited so units are presented as L m-2 and W m-2.

25. Comments from Referee: "P17 L13: Repetition of P12 L11-20, and immediately

above. Revise, avoid repetition."

Author's response: The results discussed on Pg 12 describe the maximum depths in the ice that could support photosynthesis based on maximum PAR values. In contrast, the discussion on Pg 17 emphasizes the shallowest depths in the WCA that could theoretically support photosynthesis under the minimum PAR flux values of civil twilight.

Author's changes in manuscript: None

26. Comments from Referee: "P18: L7: Again, perhaps see Stevens et al. (2018). L18: useful to consider Xiang et al. (2009, FEMS Microbiology Ecology) as this is not altogether a new concept and has been discussed in the literature. L30: "emergence of ice in the actively melting ablation area" might be a stronger phrasing."

Author's response: L7: The revision cites Stevens et al. (2018) elsewhere but since this study focused on water (and not microbial) transport, it is not as relevant to this discussion as the work of Irvine-Fynn et al. (2012), which measured concentrations and fluxes of cells from a supraglacial catchment. L18: We realize that ice-immured microbes released into meltwater is not a new or previously unrecognized concept. Our main point is to stress that the differences observed in the biogeochemical and molecular data at depths >2 m are consistent with the microbial assemblages that were directly released by melting from englacial ice. L30: We agree with this excellent suggestion.

Author's changes in manuscript: The text on Pg 19 line 5 has been edited as recommended.

27. Comments from Referee: "P19 L9: Unclear why access to bed is relevant here, suggest simply noting the fracture networks present in temperate ice may provide an explanation for near-surface to englacial linkage might be sufficient, particularly for emergent ice in the ablation area. L31: use "more than" rather than > here."

Author's response: Agreed

Author's changes in manuscript: Both suggested changes have been made (Pg 19 lines 15-16; Pg 20, line 8).

---

## Editor Decision (ED1)

**Editor's comments**

Dear authors,

I am very pleased how you addressed the reviewers' comments and I recommend publishing your manuscript subject to the following technical corrections:

1. Page3, line 20: I suggest adding degree sign " ° " after 61 and 147.
2. Page 4, lines 7-10: I am not sure what the hyphens before the numbers mean, for example, "four-30 cm diameter …"
3. Page 6, line 2: I suggest adding a reference here.
4. Page 6, line 21: "snowpack": one word
5. Page 7, line 11: the units of PAR the photosynthetically active radiation you provide seem rather strange to me as a physical scientist.
6. Page 9, line 2: I suggest using SI units for the activation energy (kJ instead of cal).
7. Page 13, line 13: I suggest replacing rho by correlation coefficient.
8. Page 21, line20. I think it is nice practice to thank the reviewers.
9. Figure 1: I do not understand why it reads "ice water content" in the caption. It seems to me only ice temperatures are shown? Moreover, the proper units are (°C). Please add degree sign for surface and ice temperature. Finally, for date format, "dd mmm", is preferred as it is unambiguous.

3 Nov 2018
*Jürg Schweizer*

---

## Author Response (AR2)

**Response to the Editor's comments**
**5 November 2018**

*1. Page3, line 20: I suggest adding degree sign " ° " after 61 and 147.*

We have inserted degree signs to these coordinates.

*2. Page 4, lines 7-10: I am not sure what the hyphens before the numbers mean, for example, "four-30 cm diameter …"*

The hyphens have been removed from these areas of the text.

*3. Page 6, line 2: I suggest adding a reference here.*

We have added a reference that provides information on the Euler central difference method.

*4. Page 6, line 21: "snowpack": one word*

The change has been made.

*5. Page 7, line 11: the units of PAR the photosynthetically active radiation you provide seem rather strange to me as a physical scientist.*

There is not an official SI unit for photosynthetic photon flux density and $\mu$mol m$^{-2}$ s$^{-1}$ is the most extensively used in photobiology and the unit recommended by the International Commission on Illumination (CIE, Terminology for photosynthetically active radiation for plants. CIE Collect Photobiol Photochem, 106, 42–46, 1993).

*6. Page 9, line 2: I suggest using SI units for the activation energy (kJ instead of cal).*

The energy of activation is now presented as kJ.

*7. Page 13, line 13: I suggest replacing rho by correlation coefficient.*

This suggested change has been made.

*8. Page 21, line20. I think it is nice practice to thank the reviewers.*

Agreed. We now do this in the acknowledgements section.

*9. Figure 1: I do not understand why it reads "ice water content" in the caption. It seems to me only ice temperatures are shown? Moreover, the proper units are (°C). Please add degree sign for surface and ice temperature. Finally, for date format, "dd mmm", is preferred as it is unambiguous.*

The meltwater volume is the red line in panel B and the text describes how this was estimated (pg. 6, lines 16-19). Degree signs have been added to panels B and C; the date format has also been changed as suggested.